## [Transparent Peer Review file · Nature Communications]

Reconstruction of the lifeways of Central European Late Bronze Age communities using ancient DNA, isotope and osteoarchaeological analyses

Corresponding Author: Dr Wolfgang Haak

Version 0:

Reviewer comments:

Reviewer #1

(Remarks to the Author)

This manuscript offers new genetic and isotopic analyses of ca. 70 LBA Central European inhumations in order to reconstruct population and mobility at the end of the second and start of the first millennia BC – a period when the dominant funerary rite was cremation, limiting our ability to conduct some biomolecular research. As such, it fills a major knowledge gap in our understanding of the population history of later prehistoric Europe. It is also innovative in its extremely thoughtful integration of genetics, isotopic data, and mortuary archaeology. I also want to note that the entire text is exceptionally well written with a crystal clear methods section and a really thoughtful discussion. I commend the authors on work of this quality. The figures are generally clear though sometimes difficult to read due to the lower resolution of reviewer pdfs. The supplementary material are complete, easily navigable and largely helpful. I was surprised not to see a discussion of the isotopic work in the supplementary material – I think this would be welcome.

All my specific comments are minor and just to help tweak and clarify the text

Abstract

Ln 54-56 “However, reconstructing the demographic and social impacts of these changes has been hindered by cremation being the dominant mortuary practice, limiting biomolecular approaches.” – this is a nitpick, but the dominant rite being cremation does not hinder our reconstruction of the *social impacts* of mobility and transformation – we can do that just fine without biomolecular data

Manuscript

Ln 81 “buried in fields” – change fields to “flat cemeteries”

Ln 99 “accompanied by stone packs” – unclear phrasing. Does this mean “stone packing”?

Ln 122 “outside the second settlement’s “gates”” – perhaps “outside the entry to the second settlement”? ‘Gates’ in scare-quotes here is confusing, especially as the figure referenced does not show much detail of either enclosure

Ln 304-05 “without being able to locate the allochthonous source concretely” – this seems in fact entirely expected within the scenario outlined by the authors of regular small-medium scale mobility and more intensive connectivity. Instead of a single or major pt A pt B geneflow event, people intermingle and become mixed with shifting patterns of ancestry resulting rather than one or more ‘sources’ of this ancestry. I’d encourage the authors to trust their data and perhaps think away from ‘origins’ to the social practices and relationships underlying their data (remembering too that all of the ‘populations’ that are being modelled in these admixture models are statistical entities not bounded, isolated groups of self-identifying people).

Ln 525 “those receiving formal burial” – are you including inhumed and cremated interments here or just inhumations? There’s a bit of slippage in this section where it’s not always clear whether you’re addressing the totality of the burial community, or the group of inhumations assessed in this manuscript.

Ln 534-36 “Despite the local adherence to inhumation practices, which might initially suggest a conservative cultural stance, the Unstrut group was connected to neighbouring, predominantly southern LBA groups” – the sentence implies that (a) retention of an older burial rite is conservative (a word with strong implications of unchanging), (b) that conservative people are not in contact with other communities – neither of these is true. Traditions and old ways are enacted because they are valuable in the present – many traditions are themselves invented or reinvented or reinvested with new meaning to address

dynamic presents. Moreover, there is no reason to assume that a group with deeper rooted or intensely local funerary rites is not in touch with neighbours or that being in touch with neighbours (even having children with them) will necessarily lead to changes in burial rite or the adoption of the outside burial rite. Underlying this sentence is an unwarranted (and probably unconscious) evolutionary logic that does not really survive being closely examined. I'd suggest a rethink or a rephrase. Maybe to something like "Even as the Unstrut group developed and maintained connections to neighbouring, predominantly southern LBA groups, they continued to practice inhumation burial, a rite with a deep history in this part of central Germany" The authors might find the discussion of parallel, distinct Neolithic Irish mortuary rites in Smythe et al (2025 - <https://doi.org/10.1017/S0959774325000058>) of interest.

Ln 549 – typo (two commas)

References

#34: this is a chapter in The Oxford Handbook of the European Bronze Age, not a book or report as it's presently cited

Figures

Fig 3b: I think the authors might experiment a bit with this image – at present, though they note in the text that data are missing from some periods, it's not visually obvious that this is the case. To make the four graphs more coherent amongst each other, I think it might be useful for them to have all the same X axis, even if some categorise (e.g. MBA Central Europe) n=z and the space is empty.

There is I think a missing figures relating to the Sr data: A broader Sr basemap so that your local values can be visually put in context with the wider region what proximity to these sites falls within the local range (e.g. if most people are travelling and living their lives on the local geology, is that within a 15 km range, a 50km range, a 200 km range?). It would be useful to indicate as well where you collected each of your baseline samples for your tightly defined local area.

Supplementary material

SM part 1: I would have liked a couple of maps to locate the cemeteries more precisely and then locate the graves within them. The photographs are extremely welcome, but a diagram of their placement would be very helpful for future researchers.

Supplementary tables: a contents tab that summarises what is in each of the sheets would be welcome but is optional

Overall, I very much enjoyed reading this manuscript! Catherine Frieman

Reviewer #2

(Remarks to the Author)

The attachment contains a formatted version of these comments.

Reviewer's comments on "Ancient DNA, isotope and osteoarchaeological 1 analyses inform on lifeways of Late Bronze Age communities in Central Europe"

Journal questions

What are the noteworthy results?

The paper presents results from period and location that is otherwise poorly represented because of the destruction of genetic material by the dominant cremation rite during the Urnfield culture. It demonstrates that cultural changes were accompanied by gradual changes in ancestry.

Will the work be of significance to the field and related fields? How does it compare to the established literature? If the work is not original, please provide relevant references.

This is a significant addition to the literature on the genetics of late bronze age Europe. It also integrates isotopic and cultural data to an extent that has previously been unusual in the ancient DNA literature.

Does the work support the conclusions and claims, or is additional evidence needed?

I consider that the data and analysis supports the conclusions, though some minor aspects of methods and results need improved presentation.

Are there any flaws in the data analysis, interpretation and conclusions? - Do these prohibit publication or require revision?

There is nothing here that I consider prohibits publication, but see the detailed comments below on some deficiencies in the analysis, and caveats on interpretation.

Is the methodology sound? Does the work meet the expected standards in your field?

Yes, the overall approach is sound and uses established methods. The methods meet expected standards, as does most of the data reporting, but see below for some omissions.

Is there enough detail provided in the methods for the work to be reproduced?

For the most part. I am not expert in the genetic methods but they are presented at the same level of detail I have seen in other papers and I am able to follow how the results were produced. Most of the isotopic and dating methods are presented fully enough, but there are some missing (see below). Statistical methods are adequately presented.

Detailed comments

Lines 200-238 use f_4 statistics of the form $f_4(\text{OldAfrica}, \text{EEF}, X, Y)$ and cite supplementary table 3, but comparisons with EEF as population2 do not appear in that table.

The tests used are “either a non-parametric Kruskal-Wallis test or a parametric ANOVA, depending on the data distribution” (lines 275-276 and 1013-1014). This may lead to incoherent results as K-W comparisons are based on medians and ANOVA on means. It would be better to use the same test throughout. Figure 3b shows one sample size is small ($n=11$) for the ANOVA, which makes deviations from normality hard to detect and suggests the use of K-W. However, the p-values here are so low this is unlikely to change the conclusions. What is the meaning of one, two and three asterisks?

Lines 365-374: When reporting ROH the units should be given, presumably cM. Supp Table 7 shows that KNE028 and KUC027 also have $\text{sum_roh} > 4$ over 50, and the former has sum_roh greater than KUC010 in all ranges, but they are not discussed here. Why?

Missing data and/or methods. The methods indicate carbon and oxygen isotope analyses were performed using enamel carbonate (lines 992-1009), but Supp Table 8 only reports $\delta^{18}\text{O}$ and not $\delta^{13}\text{C}$. Figure 6d shows $\delta^{13}\text{C}$ collagen results, but stable isotope measurements on collagen are not described in the methods, which only include collagen preparation and measurement for radiocarbon. I could not find the $\delta^{13}\text{C}$ data plotted in figure 6d in the supplementary information (“ ^{13}C ” appears in the file only in Supp Table 12), though millet consumption yes/no is in Supp Table 1. Statistical analysis of both collagen and enamel $\delta^{13}\text{C}$ data is reported (lines 480-481 and Supp Table 12). Does some of this data come from Wahl & Price (2013)? Please could $\delta^{13}\text{C}$ carbonate, $\delta^{13}\text{C}$ collagen and $\delta^{15}\text{N}$ collagen be added to supp Table 8?

Local oxygen isotope range. Lines 384-385: one cannot define a human local $\delta^{18}\text{O}$ baseline from animals as there is “marked species specificity in the relationship between drinking water and body water” (ref. 16) with a strong effect of body mass, making hamsters particularly unsuitable. The construction of this baseline is not described in the methods. The range of faunal values in Supp Table 8 is -8.3 to -5.6 ‰, but the range in Figure 6c is greater than this, from below -9 to above -5 ‰. Without data to indicate the local range, I think the best approach is to look for outliers. Here none of the values are more than 1.5xIQR from the quartiles, so no outliers are detected.

Line 418: C3 plants would be ubiquitous on all local geological formations, not found on different ones to millet, but millet may have been restricted to certain geological formations.

Lines 688-689: It would be helpful if Supp Table 1 indicated which are the new dates and which are from Orfanou et al (2024).

Line 925 “when possible, we sampled the enamel from both first and third molars” This does not seem to have been done. Supp Table 8 shows only one individual with two teeth sampled, and those were M2 and M3. Overall M2 is the most frequently sampled tooth.

Line 927 “ $\delta^{18}\text{O}$ values from M1 should be interpreted with caution as they can be influenced by breastfeeding” This is true. These samples need to be marked when plotted in figure 6, and perhaps omitted from the outlier analysis I suggested above.

Lines 990-1009: the reference points (VPDB or VSMOW) for the δ values should be stated at least once.

Supp Table 6 IBD: the units, presumably cM, should be given.

Supplementary Text

Line 83 “Dating: LBA (MAMS-57256: 941 ± 41 BC cal (Cal 1-sigma))”

Line 241 “Dating: 14C: cal BC 894–830 (Cal 1-sigma), cal BC 902–819 (Cal 2-sigma).”

Line 921 “Individual A - MAMS-52806: 3044 ± 25 BP”

The first two (and many others) are not properly reported radiocarbon dates. The third (like others in pages 47 to 49 of the supplement) is inconsistent in not supplying calendar dates. The calibrated result should be expressed as a range with a probability; following Supp Table 1 for line 83 this would be 1006-840 BC (95.4%). According to the methods “All calibrated dates are reported in the text as 95.4% probability ranges” and this should also be the convention for every date in the supplementary text. The laboratory code for the date should always be included.

Where there are multiple individuals in a grave it would be useful to know which of them the radiocarbon date comes from. Where there are multiple dates, the individuals should be identified with the relevant date (e.g. lines 498-504)

Line 113: “ 1173.5 ± 47.5 ” this is spurious precision – the calibration data is at 5 year intervals and dates cannot be

determined to a fraction of a year. Round to the whole year.

Many descriptions contain mentions of profiles A-B, C-D and E-F, but these are not marked on the images and are therefore not meaningful to the reader.

Supplementary Figure 3: what do the colours in this figure indicate?

Minor changes

Figure 4 I can't see any greyed out lines here. Are these non-working models the lighter colours?

Line 549 delete extra comma

Line 687 "95.4% (2σ) probability ranges" – delete " 2σ " these are not standard deviations.

Lines 980: ± 13 should be ± 0.000013

Line 995: NaClO should be NaClO (lower case l not uppercase I)

Line 1000: CO2 should be CO₂

Line 997: liophilized should be lyophilized

Supplementary throughout:

use point not comma for a decimal;

use raised "" not lowered „

Standardise age group names: choose one of juvenil, juvenile or juvenis; matur, mature or maturus; adult or adultus.

Supp Line 355: "right stool" – I don't know what this means

Supp Lines 498-504: remove 'mit'

Reviewer #3

(Remarks to the Author)

The paper by Eleftheria Orfanou and contributors aimed to address aDNA, provenance isotope analysis, combined with osteoarchaeological analysis, to understand the lifeways of populations from central Europe during the Late Bronze Age. Unfortunately, as I am unable to comment on some of the core aspects of the paper, such as genetic and isotope analysis, this review will focus on the osteoarchaeology/ palaeopathology/ funerary aspects of the paper.

Content and interpretation

-Lines 455-457: the authors interpret the lack of biological relationships between the individuals buried within a complex mortuary feature (39/2018) as resulting from cultural or ritual criteria instead of biological relatedness. Now, can some of the explanations be attributed to 'opportunistic or convenient burial reasons'? For instance, it would not be irrational to think that if both individuals displayed perimortem trauma that resulted from a fatal event occurring in close temporality to one another, wouldn't it be convenient to bury both (genetically unrelated) individuals within the same grave? For this reason, I suggest that, in addition to 'cultural or ritual criteria', the idea of logically opportunistic or convenient burial should be considered as an explanation.

-Lines 497-498: stated that 'these inhumations also provided us with a rare opportunity to assess the physiological stress and trauma of the people in LBA central Germany. You commence the descriptions, indicating that 'degenerative diseases were most common in older individuals' now, do you consider joint diseases to be physiologically related or trauma-related? Unless you believe that joint diseases represent 'trauma to the joints', I think that instead of referring to 'physiological stress and trauma', it would be better suited to inform the audience that you looked at palaeopathological lesions/conditions (or just the palaeopathology of the populations). These will include joint diseases, trauma, and metabolic conditions (markers of non-specific physiological stress). Also, in Line 505, you mentioned tuberculosis, which is neither a form of trauma nor a stress marker; therefore, be careful when stating that you observed the evidence for trauma and physiological stress because your analysis went beyond that and included more types of pathologies. For this reason, I suggest you use the term or talk about 'pathologies or palaeopathologies in general'

-Line 503: replace 'stress markers', which is a broad term, with 'Non-specific marker of physiological stress', which is the common term used to describe cribra orbitalia, porotic hyperostosis and dental enamel hypoplasia. Replace 'stress markers' everywhere else in the paper when you are referring to these markers and use the one suggested above.

-Line 505: what do you mean by 'spinal tuberculosis'? If you only found periosteal reactions (new bone formation on visceral aspects of ribs) without the pathognomonic signs of destructive lesions in the spine, then you cannot be certain it was TB, just a non-specific infection.

-Lines 521-525: the summary of the palaeopathological evidence (or the discussion, Lines 610-649) could have included some interpretations on the dental disease. If you were interested in the lifeways of these populations... What were the health implications associated with the findings of *Streptococcus mutans* and *Parvimonas*?

Finally, was the interpretation of body positions out of the scope of analysis? Some individuals were buried prone, which, if analysed in conjunction with isotope analysis and grave goods, could have funerary implications that could contribute to addressing the lifeways of these populations.

Typos and grammar:

-Line 549: delete one comma

-Line 676: change 'was' with 'were'

-Lines 1035-1036: accession number and URL link not provided

Reviewer #4

(Remarks to the Author)

Orfanou and colleagues present a genomic, isotopic, and osteoarchaeological analysis of rare inhumation burials from populations in Central Europe associated with the Urnfield culture, during the Late Bronze Age (LBA). The focus of the manuscript is site-specific and regional, targeting a local population present at the archaeological sites of Kuckenburg and Esperstedt in Central Germany, but they supplement with novel data derived from contemporaneous ancient individuals from South Germany, Bohemia (Czechia) and southeastern Poland. They expand the genomic analyses further with previously published contemporary and ancient human datasets. They conclude for the local Kuckenburg and Esperstedt population that there were no associations between mortuary practice and genetic sex, relatedness, geographic origin, or inferred socioeconomic status. The community showed cultural flexibility, with inhumation vs cremation behavior possibly negotiated with fluctuating ideas of identity based on external influences, as regional interconnectedness grew. This local and more nuanced view of an ancient community is commendable, the paper is generally well written and clear, and I see no deep methodological faults.

There are improvements that could be made to strengthen the manuscript. I will discuss these generally, and then provide more detailed line edits below.

The first concerns the incorporation of the new genomic data from South Germany, Czechia, and Poland. There is no background information provided for these individuals in the Introduction, and they are skipped completely some section (e.g. Mortuary practices and physiological stress), which makes them feel 'tacked-on' to the project (I am aware the burial details are in the supplemental). Apart from these individuals being contemporaneous in age and (apparently) associated with the Urnfield Culture, the reader is left unclear why these are being compared to the Unstrut group; are they the only other Urnfield populations sampled (i.e. the AADR lacks representatives for this culture, given the prevalence of cremation?). Where do these populations fit into the question presented in the introduction, of cultural and biological continuity vs change, in Central Europe, during the Bronze Age? As a more detailed example, should we have expected different ratios of EEF, Steppe, and WHG than what is estimated with the qpAdm results in Figure 2b? Are those results very surprising, from an archaeological/cultural standpoint?

The second concerns the continued theme throughout the manuscript of fluctuating Early European Farmer (EEF) related ancestry. There doesn't seem to be a consistent story told about population continuity vs admixture, but I'm sure a few sentences would clear it up. Some examples: The paper highlights a fairly consistent increase in EEF ancestry over time in all regions (Figure 3), but the qpAdm modeling suggests population continuity from EBA to early LBA (Central Germany) or continuity from MBA to LBA (Bohemia and Poland). In the strontium analyses, everyone seems local, regardless of time depth, genetic sex, or mortuary practice – where in the increase in EEF ancestry coming from? In the dietary analyses, there does seem to be a correlation with a return to C3 plants and EEF-related ancestry, in the late LBA, but not with the earlier switch to millet. The authors then bounce between suggesting a change in ancestry with the former, but not one with the latter. Settlement contexts is another example.

The authors could expand the mtDNA and NRY analyses and interpretations. The continental spread and phylogenetic history of the haplotypes typed are not discussed, even in passing. Are any of them surprising (the two CT individuals)? All expected? How about the drastically increased mitochondrial diversity? KUC023 has a unique NRY haplotype and an elevated Sr ratio, but is not apparently a genetic outlier. What might be going on there?

Some quick comments on the methods:

Was aDNA extraction attempted on any of the bones from the cremated remains? Are the authors just not reporting libraries at all that didn't pass internal endogenous DNA scan requirements? They had the bones for the isotopic work.

In your biological relatedness analyses, KIN (<https://doi.org/10.1186/s13059-023-02847-7>) seems to be growing in popularity and perhaps should be attempted here? Furthermore, is the limited relatedness detected at Kuckenburg and Esperstedt to be expected? The authors could expound on this more.

Specific Comments:

Page 3, lines 81-83: Give a date range here?

Page 5, lines 139-141: Are the "Sonderbestattungen" burials the only unique archaeological/cultural aspect of the Kuckenburg and Esperstedt settlements? Would the results presented here be applicable to the other non-cremation sites mentioned in Paragraph 2 of the introduction? How broadly applicable would these analyses be to Bronze Age central European communities, or do the authors want the takeaway to be a hyper site-specific perspective?

Page 5, lines 162-164: I can follow why NES005 was thrown away, but I'm not following why NES006 wasn't - high

contamination with angsd and hapcon, but low with ContamMix. Have you considered you might have sex chromosomal aneuploidies? Could check with karyo_RxRy? Also, if I'm following your column BW in Sup Table 1, you only threw out 2 individuals (though NES006 is in red), which seems to leave 72 individuals for other analyses; not 70. Are there duplicates? Column H has 4 "same" values and maybe that is where the difference is. If so, mention 2 duplicates were found in 4 analyses.

Page 7, lines 181-182: Which individuals (genomic data) are the authors using to define these three groups in their analyses? I can't find those group IDs in any of the supplementary tables. If they are consistent samples that everyone uses, define and give appropriate references.

Page 7, lines 183-187: This belongs more in the Methods section, than in Results

Page 7, lines 194-196 (Figure 2b Legend): An expanded introduction that includes the S. Germany, Czechia, and Poland individuals would help give us more perspective of how we should be interpreting the qpAdm models in comparison to the Unstrut population.

Page 8, lines 219-220: A description of what makes this cemetery unique likely belongs in the introduction.

Page 8, line 240: Edit to "...generated, as well as published, LBA..."

Page 9, lines 256-258: I don't understand where the support for this sentence comes from in the preceding paragraph, as I don't see any graphics or data in Supp Table 3 showing any early bronze age comparisons?

Page 9, lines 260-269: Why are the individuals from Poland, who seem to have the lowest proportion of EEF-related ancestry, not discussed in this paragraph?

Page 9, lines 271-274: Can you be more clear where you are getting the proportions used for Figure 3a/b, because they don't seem to exactly match the values from qpAdm listed in Figure 2b (for the LBA individuals, at least). This can be put in the text or in the figure legend.

Pages 12-13, lines 333-342: Also discuss LNV001 and SNY001, who are highlighted as outliers in this manuscript, in the PCA, and in Figure 5 below.

Page 14, lines 365-374: Why are KNE028, KUC027 not being discussed in the hapROH analysis? Do they not also have elevated levels? See Supplemental Figure 3.

Page 14, lines 386-396: None of the genetic outliers appear non-local in the isotopic analyses. Can the authors discuss/highlight this?

Page 15, line 406: How as millet consumption determined? I see the column in Sup Table 1, but not a description.

Page 21, line 571: By "this dietary shift" I assume the authors are referring to the return C3 plants, but please clarify.

Page 27, line 730: The 1240k capture needs references; I assume you aren't using the new Twist kit.

Page 27, lines 733-736: Both the mtDNA and NRY capture protocols/kits need references.

Page 30, line 804: I assume these genotypes are from the preceding ATLAS calls? Please be more specific in the "Population Genetic Analyses" when you used the pseudo-haploid calls and when you used the ATLAS genotypes.

Page 30, line 807: spelling "pubblication"

Page 31, lines 831-833: Was the HO dataset *only* used for the PCA and Admixture analyses? If nothing else, say that. I would also state the limited HO dataset was necessary to include many of the contemporary populations.

Page 31, line 853: Here and elsewhere, when you bring up population group labels like this, reference your Supplementary Table 2b.

Page 31, lines 856-857: As already previously mentioned, we are never given who are included in these groups.

Page 32, lines 867-872: Are these labels from the AADR or from labels in Sup Table 2b? I don't see some of them in the supplementary tables.

Page 37, lines 1031-1032: I applaud the authors for including this level of detail for their statistical tests.

Reviewer comments:

Reviewer #1

(Remarks to the Author)

I was really pleased to see the careful and thoughtful revisions and responses to the reviewers. The authors have made an already strong paper excellent. I have no further comments - well done!

-Catherine Frieman

Reviewer #2

(Remarks to the Author)

The authors have provided a comprehensive reply to the comments and have made improvements to their manuscript. I have only one further comment: the reply makes clear the source of the collagen isotopic data. For complete clarity I think they ought to cite Orfanou et al. (2024) as the source in the caption for Figure 6d.

Reviewer #3

(Remarks to the Author)

I have reviewed the manuscript for the second time, and in my opinion, the authors satisfactorily modified the text in accordance with my former suggestions. I appreciate the time and effort they have put into acknowledging these. Thus, I see no reason why the manuscript could be published.

Reviewer #4

(Remarks to the Author)

I appreciate the authors for responding positively and meticulously to most of my suggestions. I now read a more nuanced and regional take on the EEF ancestry changes, more highlights of when results matched or didn't match expectations, and expanded discussions of the diet correlations and relatedness measurements. More specifically, thanks to the authors for running KIN, for reporting on the single cremation they attempted aDNA analysis on, for the new supplemental note on KUC023, for clarifying which genotype sets were used when, and for providing much more detail on ancestry group names and membership, both in the text and in supplemental tables – this makes future replication easier and our field better.

(All subsequent line references are to the tracked changes pdf)

In regards to the individuals genotyped from South Germany, Bohemia, and SW/Central Poland, I acknowledge the broader genetic background of Bronze Age Central Europeans given in the Introduction and I acknowledge that the authors more consistently highlight these individuals in the Results. I am still unclear why, however, these individuals are only being introduced to us halfway through the Results, particularly when most of the first few paragraphs of that section incorporate them as part of a larger Central European LBA gene pool and they are identified in Figure 1. While I understand the focus is on the Esperstedt and Kuckenbug Unstrut group, would it truly disorganize the manuscript to summarize those 3 additional populations in a minimized version of what you do in lines 132-159 for your main site(s)? You describe as much in your peer review rebuttal notes. Instead, we still only get a single sentence heads up, in the final paragraph of the introduction, of their existence, even though they represent almost half of your (newly contributed) dataset. It is the authors' prerogative to bury (ha) or to highlight sites, but I do think it is a missed opportunity. If LBA individuals from Central Europe are universally scarce, shouldn't any and all be meaningful?

Some new small edits:

In regards to KUC023, fix the reference at line 190 to say 'Supplementary Note 2.4' instead of 'Supplementary Table 2.4', and add a reference to Supplemental Note 2.4 somewhere in lines 433-436.

Additionally include a short note of KUC027 and KUC010 in Supplementary Note 2.3, as you have already done in the main body of the text. That is where the reader will see the actual results.

Thank you for clarifying on final sample counts (69 passing quality control), though your edits on lines 186-187 now says "51 different mtDNA haplotypes in 64 individuals", and I'm unclear why.

There are some changes in font style at lines 944-946 and 1084-1086.

Point-by-point response to the and Reviewers

.....

Reviewer 1

This manuscript offers new genetic and isotopic analyses of ca. 70 LBA Central European inhumations in order to reconstruct population and mobility at the end of the second and start of the first millennia BC – a period when the dominant funerary rite was cremation, limiting our ability to conduct some biomolecular research. As such, it fills a major knowledge gap in our understanding of the population history of later prehistoric Europe. It is also innovative in its extremely thoughtful integration of genetics, isotopic data, and mortuary archaeology. I also want to note that the entire text is exceptionally well written with a crystal clear methods section and a really thoughtful discussion. I commend the authors on work of this quality. The figures are generally clear though sometimes difficult to read due to the lower resolution of reviewer pdfs. The supplementary material are complete, easily navigable and largely helpful.

We sincerely thank Reviewer 1 for the positive and constructive evaluation of our manuscript. Regarding the embedded figures, we have ensured that the final figures are provided at a higher level of resolution.

I was surprised not to see a discussion of the isotopic work in the supplementary material – I think this would be welcome.

Supplementary Note 3 should have been named *isotope analyses* instead of *statistical analyses*, since it contains statistical analyses of isotope data. This has now been corrected. Furthermore, we have now expanded this section with a paragraph regarding

$\delta^{18}\text{O}$ outlier detection (1.5×IQR method) in human enamel (Supplementary Note 3.3) and a section on the regional $^{87}\text{Sr}/^{86}\text{Sr}$ isoscape context (Supplementary Note 3.2).

Abstract

Ln 54-56 “However, reconstructing the demographic and social impacts of these changes has been hindered by cremation being the dominant mortuary practice, limiting biomolecular approaches.” – this is a nitpick, but the dominant rite being cremation does not hinder our reconstruction of the *social impacts* of mobility and transformation – we can do that just fine without biomolecular data.

We agree that the dominance of cremation does not hinder archaeological reconstructions of social impact. We have therefore revised the sentence to refer specifically to the demographic aspects accessible through biomolecular methods. The sentence now reads: “*However, reconstructing the demographic aspects of these changes has been hindered by cremation being the dominant mortuary practice, limiting biomolecular approaches.*”

Manuscript

Ln 81 “buried in fields” – change fields to “flat cemeteries”

We have changed this in the text.

Ln 99 “accompanied by stone packs” – unclear phrasing. Does this mean “stone packing”?

We have changed this in the text.

Ln 122 “outside the second settlement’s ‘gates’” – perhaps “outside the entry to the second settlement”? ‘Gates’ in scare-quotes here is confusing, especially as the figure referenced does not show much detail of either enclosure.

We have replaced “outside the second settlement’s ‘gates’” with “*outside the ditch bordering the settlement to the west*” to improve clarity and avoid potential confusion.

Ln 304-05 “without being able to locate the allochthonous source concretely” – this seems in fact entirely expected within the scenario outlined by the authors of regular small-medium scale mobility and more intensive connectivity. Instead of a single or major pt A pt B geneflow event, people intermingle and become mixed with shifting patterns of ancestry resulting rather than one or more ‘sources’ of this ancestry. I’d encourage the authors to trust their data and perhaps think away from ‘origins’ to the social practices and relationships underlying their data (remembering too that all of the ‘populations’ that are being modelled in these admixture models are statistical entities not bounded, isolated groups of self-identifying people).

We fully agree that our results are best understood in the context of ongoing interaction and connectivity rather than discrete population movements. We have revised the phrasing to avoid implying a single allochthonous source, replacing “without being able to locate the allochthonous source concretely” with “*without being able to be more specific*”.

Ln 525 “those receiving formal burial” – are you including inhumed and cremated interments here or just inhumations? There’s a bit of slippage in this section where it’s not always clear whether you’re addressing the totality of the burial community, or the group of inhumations assessed in this manuscript.

In this section, we are indeed referring to the group of inhumations assessed in this study. We have revised the sentence to read: “*at least for the inhumed individuals receiving formal burial*” to make this distinction clearer.

Ln 534-36 “Despite the local adherence to inhumation practices, which might initially suggest a conservative cultural stance, the Unstrut group was connected to neighbouring, predominantly southern LBA groups” – the sentence implies that (a) retention of an older burial rite is conservative (a word with strong implications of unchanging), (b) that conservative people are not in contact with other communities – neither of these is true. Traditions and old ways are enacted because they are valuable in the present – many traditions are themselves invented or reinvented or reinvested with new meaning to address dynamic presents. Moreover, there is no reason to assume that a group with deeper rooted or intensely local funerary rites is not in touch with neighbours or that being in touch with neighbours (even having children with them) will necessarily lead to changes in burial rite or the adoption of the outside burial rite. Underlying this sentence is an unwarranted (and probably unconscious) evolutionary logic that does not really survive being closely examined. I’d suggest a rethink or a rephrase. Maybe to something like “Even as the Unstrut group developed and maintained connections to neighbouring, predominantly southern LBA groups, they continued to practice inhumation burial, a rite with a deep history in this part of central Germany” The authors might find the discussion of parallel, distinct Neolithic Irish mortuary rites in Smythe et al (2025 - <https://doi.org/10.1017/S0959774325000058>) of interest.

Thank you for this comment. We agree that our original phrasing unintentionally implied an evolutionary logic that is neither accurate nor theoretically appropriate. We have now revised the sentence, as suggested, to avoid these implications. The text now reads: “*Even as the Unstrut group developed and maintained connections to neighbouring, predominantly southern LBA groups, as evidenced also by the continuous increase in EEF-like ancestry, they continued to practice inhumation burial, a rite with a deep history in this part of Central Germany.*” Thank you as well for directing us to Carlin N, Smyth J, Frieman CJ, et al (2025). We found their interpretive framework very helpful and

interesting, and we now cite this work in our discussion when addressing limited biological relatedness in prehistoric mortuary contexts.

Ln 549 – typo (two commas) Done.

References

#34: This is a chapter in *The Oxford Handbook of the European Bronze Age*, not a book or report as it's presently cited

The reference has been corrected to indicate that this work is a chapter within *The Oxford Handbook of the European Bronze Age*.

Figures

Fig 3b: I think the authors might experiment a bit with this image – at present, though they note in the text that data are missing from some periods, it's not visually obvious that this is the case. To make the four graphs more coherent amongst each other, I think it might be useful for them to have all the same X axis, even if some categories (e.g. MBA Central Europe) n=z and the space is empty.

We have updated Figure 3b accordingly.

There is I think a missing figures relating to the Sr data: A broader Sr basemap so that your local values can be visually put in context with the wider region what proximity to these sites falls within the local range (e.g. if most people are travelling and living their lives on the local geology, is that within a 15 km range, a 50km range, a 200 km range?). It would be useful to indicate as well where you collected each of your baseline samples for your tightly defined local area.

We agree that it is very useful to visualise where our sites are located in relation to the surrounding strontium isotope values. For this reason, we have now included, in the *Supplementary Text 3.2*, a map showing the modelled bioavailable $^{87}\text{Sr}/^{86}\text{Sr}$ isoscape for Central Europe, adapted from Bataille et al. (2020). On this map, we indicate the locations of our sites (Kuckenbug and Esperstedt), from which the human and faunal individuals used for the analyses were collected, as well as the site of Neckarsulm, for which published strontium isotope data are available. This addition will allow readers to broadly situate our measured values within the wider regional context. We also note, as discussed in the *Supplementary Text 3.2*, that equifinality (i.e., the occurrence of similar $^{87}\text{Sr}/^{86}\text{Sr}$ ratios in geologically distinct regions) must be considered when interpreting such data. A new figure illustrating this information has been added as *Supplementary Figure 7a and b*. We have also added a sentence in the main text directing the reader to the *Supplementary Text*, lines that reads as “*We additionally provide a modelled bioavailable $^{87}\text{Sr}/^{86}\text{Sr}$ isoscape for Central Europe (Supplementary Text 3.2, Supplementary Fig. 7a,b), adapted from Bataille et al. (2020), to place these values within a broader regional context.*”

Supplementary material

SM part 1: I would have liked a couple of maps to locate the cemeteries more precisely and then locate the graves within them. The photographs are extremely welcome, but a diagram of their placement would be very helpful for future researchers.

An overview of all archaeological sites, including Kuckenburg and Esperstedt, is already provided in Figure 1. However, we agree that a more detailed map showing the precise locations of the burials is necessary. For this reason, we have now added detailed maps of the location of the Kuckenburg (Supplementary Fig.2) and Esperstedt (Supplementary Fig.1) burials in Supplementary Note 1, under the respective site descriptions, as these two sites form the core of our study.

Supplementary tables: a contents tab that summarises what is in each of the sheets would be welcome but is optional.

We have now added a contents tab at the beginning of the Supplementary Tables file, summarising the content of each sheet.

Overall, I very much enjoyed reading this manuscript!

Thank you!

.....

Reviewer 2

The paper presents results from period and location that is otherwise poorly represented because of the destruction of genetic material by the dominant cremation rite during the Urnfield culture. It demonstrates that cultural changes were accompanied by gradual changes in ancestry. This is a significant addition to the literature on the genetics of late bronze age Europe. It also integrates isotopic and cultural data to an extent that has previously been unusual in the ancient DNA literature. I consider that the data and analysis supports the conclusions, though some minor aspects of methods and results need improved presentation. There is nothing here that I consider prohibits publication, but see the detailed comments below on some deficiencies in the analysis, and caveats on interpretation. Yes, the overall approach is sound and uses established methods. The methods meet expected standards, as does most of the data reporting, but see below for some omissions. For the most part. I am not expert in the genetic methods but they are presented at the same level of detail I have seen in other papers and I am able to follow how the results were produced. Most of the isotopic and dating methods are presented fully enough, but there are some missing (see below). Statistical methods are adequately presented.

We sincerely thank Reviewer 2 for their positive assessment of our study.

Detailed comments

Lines 200-238 use f4 statistics of the form $f4(\text{OldAfrica}, \text{EEF}, X, Y)$ and cite supplementary table 3, but comparisons with EEF as population2 do not appear in that table.

We thank the reviewer for pointing out this omission. The comparisons with EEF were indeed included in the analyses, but the label was missing from Supplementary Table 3. We have now added the EEF label to the table to clarify this.

The tests used are “either a non-parametric Kruskal-Wallis test or a parametric ANOVA, depending on the data distribution” (lines 275-276 and 1013-1014). This may lead to incoherent results as K-W comparisons are based on medians and ANOVA on means. It would be better to use the same test throughout. Figure 3b shows one sample size is small (n=11) for the ANOVA, which makes deviations from normality hard to detect and suggests the use of K-W. However, the p-values here are so low this is unlikely to change the conclusions. What is the meaning of one, two and three asterisks?

Thank you for your comment.

We fully agree. In fact, while double-checking the data we noticed data from 7 unpublished EBA individuals from Central Germany which are reserved for a separate study. After removing them we applied the same Kruskal-Wallis test to all regions. We have also clarified in the Figure 3b caption that one, two, three, and four asterisks

indicate significance levels of $p < 0.05$, $p < 0.01$, $p < 0.001$, and $p < 0.0001$, respectively.

Lines 365-374: When reporting ROH, the units should be given, presumably cM. Supp Table 7 shows that KNE028 and KUC027 also have sum_roh>4 over 50, and the former has sum_roh greater than KUC010 in all ranges, but they are not discussed here. Why?

We have now included KUC027 and KNE028 in the discussion of individuals with elevated ROH values. Both display moderate ROH levels (sum_roh > 4 = 83.72 cM and 83.89 cM, respectively), comparable to KUC010 (72.68 cM). In addition, we have specified the units (cM) for all ROH values in the text and added this information to the caption of Supplementary Table 7.

Missing data and/or methods

The methods indicate carbon and oxygen isotope analyses were performed using enamel carbonate (lines 992-1009), but Supp Table 8 only reports $\delta^{18}\text{O}$ and not $\delta^{13}\text{C}$.

The $\delta^{13}\text{C}$ results were not originally included because they were published in Orfanou et al. (2024). We agree that presenting them here improves completeness and comparability. We have therefore added the $\delta^{13}\text{C}$ values as an additional column next to the $\delta^{18}\text{C}$ results in Supplementary Table 8.

Figure 6d shows $\delta^{13}\text{C}$ collagen results, but stable isotope measurements on collagen are not described in the methods, which only include collagen preparation and measurement for radiocarbon. I could not find the $\delta^{13}\text{C}$ data plotted in Figure 6d in the supplementary information (“ ^{13}C ” appears in the file only in Supp Table 12), though millet consumption yes/no is in Supp Table 1.

Statistical analysis of both collagen and enamel $\delta^{13}\text{C}$ data is reported (lines 480-481 and Supp Table 12). Does some of this data come from Wahl & Price (2013)? Please could $\delta^{13}\text{C}$ carbonate, $\delta^{13}\text{C}$ collagen and $\delta^{15}\text{N}$ collagen be added to supp Table 8?

The $\delta^{13}\text{C}$ _collagen results plotted in Figure 6d were published in Orfanou et al. (2024), where all isotopic measurements are reported in Supplementary Table S2 (*Isotopes_Overview*). The methods for collagen extraction and stable isotope measurement are also described in detail in that publication. To facilitate cross-referencing and improve accessibility, we have now added the $\delta^{13}\text{C}$ and $\delta^{15}\text{N}$ collagen values as new columns in Supplementary Table 8 of the present study. All requested data ($\delta^{13}\text{C}$ carbonate, $\delta^{13}\text{C}$ collagen, and $\delta^{15}\text{N}$ collagen) are now summarised in Supplementary Table 8.

Local oxygen isotope range. Lines 384-385: one cannot define a human local $\delta^{18}\text{O}$ baseline from animals as there is “marked species specificity in the relationship between drinking water and body water” (ref. 16) with a strong effect of body mass, making hamsters particularly unsuitable. The construction of this baseline is not described in the methods. The range of faunal values in Supp Table 8 is -8.3 to -5.6 ‰, but the range in Figure 6c is greater than this, from below -9 to above -5 ‰. Without data to indicate the local range, I think the best approach is to look for outliers. Here, none of the values are more than 1.5xIQR from the quartiles, so no outliers are detected.

We agree that a human $\delta^{18}\text{O}$ baseline should not be derived from faunal values for the reasons explained above. Following the reviewer’s recommendation, we applied a non-parametric 1.5xIQR approach to the human $\delta^{18}\text{O}$ dataset to identify potential non-local individuals. In this test, we excluded individuals whose sampled tooth was an M1, as their values may be affected by breastfeeding. These individuals are still shown in Figure 6c, where they are labelled with “M1”. No $\delta^{18}\text{O}$ outliers were detected based on this approach. The results of the test have been added to Supplementary Table 12 under “4. $\delta^{18}\text{O}$ IQR summary (humans, excluding M1)”, and a paragraph explaining the process has been added in Supplementary Text under section “3.3 $\delta^{18}\text{O}$ outlier detection (1.5xIQR method) in human enamel”. We have also removed the oxygen baseline from Figure 6c and corrected the figure and caption. Finally, we removed the corresponding sentence referring to the oxygen baseline from lines 384 - 385.

Line 418: C3 plants would be ubiquitous on all local geological formations, not found on different ones to millet, but millet may have been restricted to certain geological formations.

We agree with this clarification, and we have now revised the sentence to read “*Millet consumers displayed higher $^{87}\text{Sr}/^{86}\text{Sr}$ values, which may suggest that millet was grown in areas with different underlying geology, whereas C₃ plants would have been ubiquitous across the local landscape*”.

Lines 688-689: It would be helpful if Supp Table 1 indicated which are the new dates and which are from Orfanou et al (2024).

We have now added an additional column, *Date_Source*, after the *Date_Note* column in Supplementary Table 1, indicating which radiocarbon dates were produced in our previous study (Orfanou et al., 2024) and which were generated for the present study.

Line 925 “when possible, we sampled the enamel from both first and third molars” This does not seem to have been done. Supp Table 8 shows only one individual with two teeth sampled, and those were M2 and M3. Overall M2 is the most frequently sampled tooth.

Our initial aim was indeed to sample enamel from both first and third molars to assess potential movement across different life stages. However, due to the limited preservation and availability of these teeth, we predominantly sampled second molars (M2). We have clarified this in the text and revised the sentence accordingly, which now reads “*For the inhumations, our initial aim was to sample enamel from both first and third molars to assess potential movement across different life stages (childhood vs. adolescence). However, due to the limited preservation and availability of these teeth, we predominantly sampled second molars (M2), which provide information about early to mid-childhood*”.

Line 927 “ $\delta^{18}\text{O}$ values from M1 should be interpreted with caution as they can be influenced by breastfeeding” This is true. These samples need to be marked when plotted in figure 6, and perhaps omitted from the outlier analysis I suggested above.

We have marked the samples in Figure 6c as “M1”, and we have excluded them from the outlier analysis.

Lines 990-1009: the reference points (VPDB or VSMOW) for the δ values should be stated at least once.

They are now stated in the text “ Final stable carbon ($\delta^{13}\text{C}$, reported relative to VPDB) and oxygen ($\delta^{18}\text{O}$, reported relative to VSMOW) isotope values ...”

Supp Table 6 IBD: the units, presumably cM, should be given.

We have now specified in the caption of Supplementary Table 6 that all IBD segment lengths are reported in centimorgans (cM).

Supplementary Text

Line 83 “Dating: LBA (MAMS-57256: 941 ± 41 BC cal (Cal 1-sigma))”

Corrected.

Line 241 “Dating: 14C: cal BC 894–830 (Cal 1-sigma), cal BC 902–819 (Cal 2-sigma).”

Corrected.

Line 921 “Individual A - MAMS-52806: 3044 ± 25 BP”

The first two (and many others) are not properly reported radiocarbon dates. The third (like others in pages 47 to 49 of the supplement) is inconsistent in not supplying calendar dates. The calibrated result should be expressed as a range

with a probability; following Supp Table 1 for line 83, this would be 1006-840 BC (95.4%). According to the methods, “All calibrated dates are reported in the text as 95.4% probability ranges”, and this should also be the convention for every date in the supplementary text. The laboratory code for the date should always be included.

Corrected.

Where there are multiple individuals in a grave, it would be useful to know which of them the radiocarbon date comes from. Where there are multiple dates, the individuals should be identified with the relevant date (e.g. lines 498-504)

We specified this in the Supplementary Text.

Line 113: “1173,5 ± 47,5” this is spurious precision – the calibration data is at 5 year intervals, and dates cannot be determined to a fraction of a year. Round to the whole year.

Corrected.

Many descriptions contain mentions of profiles A-B, C-D and E-F, but these are not marked on the images and are therefore not meaningful to the reader.

Removed from the text since they are not informative.

Supplementary Figure 3: What do the colours in this figure indicate?

The coloured bars represent different ROH length bins: dark blue = 4 - 8 cM, light blue = 8 - 12 cM, yellow = 12 - 20 cM, and red = > 20 cM. We have now included this information in the figure caption.

Minor changes

Figure 4: I can't see any greyed out lines here. Are these non-working models the lighter colours?

The non-working models are indeed indicated by lighter-coloured bars rather than greyed-out ones. We have updated the figure caption accordingly to clarify this.

Line 549: delete extra comma

Deleted.

Line 687: “95.4% (2σ) probability ranges” – delete “2σ” these are not standard deviations.

Deleted.

Lines 980: ± 13 should be ± 0.000013

Corrected.

Line 995: NaClO should be NaClO (lower case l not uppercase I)

Thanks. Done.

Line 1000: CO2 should be CO₂

We have corrected the chemical formula to display CO₂ with a subscript in the revised manuscript.

Line 997: liophilized should be lyophilized

Done.

**Supplementary throughout:
use point not comma for a decimal;**

We have replaced commas with points as decimal separators throughout the manuscript.

use raised “” not lowered „“

We have replaced the German-style quotation marks (, “) with English-style raised quotation marks (“ ”) throughout the manuscript.

Standardise age group names: choose one of juvenil, juvenile or juvenis; matur, mature or matusus; adult or adultus.

We have standardised the age group terminology throughout the main text, supplementary text, and supplementary tables, using the categories *infant*, *juvenile*, *adult*, *mature*, and *senile*.

Supp Line 355: “right stool” – I don’t know what this means

The term “right stool” was a mistranslation of the German “*rechte Hockerbestattung*”, meaning an individual buried in a right-crouched (flexed) position. We have now corrected the text accordingly.

Supp Lines 498-504: remove ‘mit’

Done.

Reviewer 3

The paper by Eleftheria Orfanou and contributors aimed to address aDNA, provenance isotope analysis, combined with osteoarchaeological analysis, to understand the lifeways of populations from central Europe during the Late Bronze Age. Unfortunately, as I am unable to comment on some of the core aspects of the paper, such as genetic and isotope analysis, this review will focus on the osteoarchaeology/palaeopathology/funerary aspects of the paper.

We thank Reviewer 3 for their expertise on the osteoarchaeological, palaeopathological, and funerary aspects of our study.

Content and interpretation

-Lines 455-457: the authors interpret the lack of biological relationships between the individuals buried within a complex mortuary feature (39/2018) as resulting from cultural or ritual criteria instead of biological relatedness. Now, can some of the explanations be attributed to ‘opportunistic or convenient burial reasons’? For instance, it would not be irrational to think that if both individuals displayed perimortem trauma that resulted from a fatal event occurring in close temporality to one another, wouldn’t it be convenient to bury both (genetically unrelated) individuals within the same grave? For this reason, I suggest that, in addition to ‘cultural or ritual criteria’, the idea of logically opportunistic or convenient burial should be considered as an explanation.

We agree that opportunistic practices should also be considered as a possible explanation. For this reason, the text now reads “*cultural, ritual, or opportunistic criteria*” to reflect that such practical considerations may have influenced the joint deposition of these individuals.

-Lines 497-498: stated that ‘these inhumations also provided us with a rare opportunity to assess the physiological stress and trauma of the people in LBA central Germany. You commence the descriptions, indicating that ‘degenerative diseases were most common in older individuals’ now, do you consider joint diseases to be physiologically related or trauma-related? Unless you believe that joint diseases represent ‘trauma to the joints’, I think that instead of referring to ‘physiological stress and trauma’, it would be better suited to inform the audience that you looked at palaeopathological lesions/conditions (or just the palaeopathology of the populations). These will include joint diseases, trauma, and metabolic conditions (markers of non-specific physiological stress). Also, in Line 505, you mentioned tuberculosis, which is neither a form of trauma nor a stress marker; therefore, be careful when stating that you observed the evidence for trauma and physiological stress because your analysis went beyond that and included more types of pathologies. For this reason, I suggest you use the term or talk about ‘pathologies or palaeopathologies in general’

We agree with the reviewer's suggestion, and we have revised the terminology to better reflect the scope of our analysis. We have now changed the section title to "*Mortuary practices and palaeopathology in LBA Central Germany.*" We have also replaced "physiological stress and trauma" with "palaeopathological lesions/conditions," and the phrase now reads as: "*These inhumation burials also provided us with a rare opportunity to assess the palaeopathological lesions/conditions of the people in LBA Central Germany.*" This revised version reflects more accurately the different categories of pathological evidence considered in our study.

-Line 503: replace 'stress markers', which is a broad term, with 'Non-specific marker of physiological stress', which is the common term used to describe cribra orbitalia, porotic hyperostosis and dental enamel hypoplasia. Replace 'stress markers' everywhere else in the paper when you are referring to these markers and use the one suggested above.

We have replaced the term "stress markers" with "non-specific markers of physiological stress" throughout the manuscript.

-Line 505: what do you mean by 'spinal tuberculosis'? If you only found periosteal reactions (new bone formation on visceral aspects of ribs) without the pathognomonic signs of destructive lesions in the spine, then you cannot be certain it was TB, just a non-specific infection.

We identified several individuals showing lesions consistent with possible or early-stage spinal tuberculosis, manifesting as intraosseous, round or oval cavities within the vertebral bodies. Definite evidence of tuberculosis is observed in individual KUC012 (16_2009), where similar lesions also occur in other skeletal elements. This evidence is presented in the Supplementary Table 13 and includes mild spondylitis of the vertebral bodies T4-L2, osteitis of the tibia, periostitis of the tibiae, among other changes. We now direct the reader to the Supplementary Table 13 by adding the following in the main text: "*Non-adults have also a higher incidence of spinal tuberculosis (e.g., individual KUC012, Supplementary Table 13)*".

-Lines 521-525: the summary of the palaeopathological evidence (or the discussion, Lines 610-649) could have included some interpretations on the dental disease. If you were interested in the lifeways of these populations... What were the health implications associated with the findings of *Streptococcus mutans* and *Parvimonas*?

We thank the reviewer for this suggestion. We did not analyse in depth the dental status and disease-related changes to the teeth and gums of the individual skeletal remains in this study. As part of the investigation, however, we did document the dental status and recorded notable pathologies for completeness. A more detailed assessment of dental

health lies beyond the scope of the present study but will be addressed in future work, once more individuals become available. What we can say from the data available to date is that, for the burials where dental status could be assessed, the individuals generally show good dental health. Calculus and mild caries are commonly observed, while isolated cases of severe caries also occur, ranging from *pulpa aperta* to pathological changes in the jawbone (e.g., KUC004). However, such severe cases are rare within the examined population. We added this observation to the text, and now it reads as *“In addition, for the burials where dental status could be assessed, the individuals generally showed good dental health. Calculus and mild caries were commonly observed, while isolated cases of severe caries also occurred, ranging from pulpa aperta to pathological changes in the jawbone (e.g., individual KUC004, Supplementary Table 13). However, such severe cases were rare within the examined population”*.

Regarding the health implications associated with the findings of *S. mutans* and *P. micra* it is difficult to draw direct conclusions. Both pathobionts are commonly found in the oral microbiome where they can cause caries and periodontitis. However, this can be prevented by good hygiene. On the other hand, it can be favoured by a carbohydrate rich diet. Detecting both pathogens, in particular *S. mutans*, is in so far no surprise, as caries has been observed in some individuals. We therefore added: *“These pathobionts are commonly involved in the development of dental caries and periodontitis, respectively, often favoured by a carbohydrate rich diet. Their detection is in line with the paleopathological observations”*.

Finally, was the interpretation of body positions out of the scope of analysis? Some individuals were buried prone, which, if analysed in conjunction with isotope analysis and grave goods, could have funerary implications that could contribute to addressing the lifeways of these populations.

We agree with the reviewer that the body positions of the individuals could provide valuable insights when considered alongside isotopic data and grave goods. However, in our current dataset, the variation in body positions is high, preventing the formation of statistically meaningful groups for analysis. Moreover, at this stage, only a few individuals are associated with grave goods. For these reasons, we did not pursue an in-depth interpretation of body positions in this study. We recognise the potential significance of this aspect and plan to explore it further in future research, when more individuals become available.

Typos and grammar:

-Line 549: delete one comma

Done.

-Line 676: change ‘was’ with ‘were’

Done.

-Lines 1035-1036: accession number and URL link not provided

We added the accession number: PRJEB98508 and the link will go live upon acceptance.

Reviewer 4

Orfanou and colleagues present a genomic, isotopic, and osteoarchaeological analysis of rare inhumation burials from populations in Central Europe associated with the Urnfield culture, during the Late Bronze Age (LBA). The focus of the manuscript is site-specific and regional, targeting a local population present at the archaeological sites of Kuckenburg and Esperstedt in Central Germany, but they supplement with novel data derived from contemporaneous ancient individuals from South Germany, Bohemia (Czechia) and southeastern Poland. They expand the genomic analyses further with previously published contemporary and ancient human datasets. They conclude for the local Kuckenburg and Esperstedt population that there were no associations between mortuary practice and genetic sex, relatedness, geographic origin, or inferred socioeconomic status. The community showed cultural flexibility, with inhumation vs cremation behavior possibly negotiated with fluctuating ideas of identity based on external influences, as regional interconnectedness grew. This local and more nuanced view of an ancient community is commendable, the paper is generally well written and clear, and I see no deep methodological faults. There are improvements that could be made to strengthen the manuscript. I will discuss these generally, and then provide more detailed line edits below.

We sincerely thank Reviewer 4 for their positive and thoughtful evaluation of our manuscript.

The first concerns the incorporation of the new genomic data from South Germany, Czechia, and Poland. There is no background information provided for these individuals in the Introduction, and they are skipped completely some section (e.g. Mortuary practices and physiological stress), which makes them feel ‘tacked-on’ to the project (I am aware the burial details are in the supplemental). Apart from these individuals being contemporaneous in age and (apparently) associated with the Urnfield Culture, the reader is left unclear why these are being compared to the Unstrut group; are they the only other Urnfield populations sampled (i.e. the AADR lacks representatives for this culture, given the prevalence of cremation)?

Our study centres on regions in today’s Central Germany, where inhumations are present during the LBA, while cremation was the predominant mortuary practice. Because of this mortuary practice, genetic data from South Germany, Czechia, and Poland are also scarce. Due to this scarcity of LBA individuals from Central Europe in public repositories (e.g. POSEIDON or AADR), aside from the few published we rely heavily on the newly generated genetic data from these neighbouring regions to situate the LBA Unstrut group within a wider Central European chronological context. The individuals associated with the Knovíz culture in Czechia date to the LBA/Urnfield period and share broadly comparable mortuary practices with Central Germany, as we mention in the Introduction. Even though published Knovíz data exist, our new

individuals provide additional, regionally relevant genetic data from the LBA. The four LBA/Lusatian/Urnfield individuals from Poland are also used for genetic comparison only. They are included because they are contemporaneous, and reasonable geographical proxies given the cultural and demographic interconnectedness of the period. As such, the combined genomic data from today's Germany, Czechia and Poland presents the first solid LBA genetic record for Central Europe and thus a useful baseline for future studies that wish to integrate multi-proxy analyses across wider regions in Europe. We have added clarifying sentences in the main text, emphasising that the southern German, Czech, and Polish individuals are used exclusively as a genetic comparative dataset. The text now reads as “ *We adopt a site-specific perspective (i.e., focusing on the sites of Esperstedt and Kuckenburg), where we reconstruct individual bioarchaeological profiles, and we contextualise their genetic ancestry through comparison of the newly analysed individuals with published contemporaneous groups from Central and South Germany, Czechia, and Poland (Fig. 1b)*”.

Where do these populations fit into the question presented in the introduction, of cultural and biological continuity vs change, in Central Europe, during the Bronze Age? As a more detailed example, should we have expected different ratios of EEF, Steppe, and WHG than what is estimated with the qpAdm results in Figure 2b? Are those results very surprising, from an archaeological/cultural standpoint?

From an archaeogenetic perspective, the proportions of EEF-, Steppe- and WHG-related ancestry observed in our LBA individuals are broadly consistent with expectations, i.e. what is known from the reported genomic ancestry profiles of individuals dating to the Bronze Age period in Central Europe, and in fact across most regions of Europe more generally (Mathieson et al. 2018, Olalde et al. 2019, Furtwängler et al. 2020, Papac et al. 2021, Patterson et al. 2021, Lazaridis et al. 2024, to name but a few).

Specifically, for South Germany, elevated EEF ancestry relative to Central Germany has been documented since the Early Bronze Age (Mittnik et al. 2019; Furtwängler et al. 2020, Gretzinger et al. 2024), and our results further confirm this observation, as mentioned in lines 292 – 305 in the main text. In Bohemia, the shift from the EBA/Únětice to the MBA/Tumulus and LBA/Knovíz cultures is also associated with higher EEF ancestry, in line with previous studies (Patterson et al. 2021), as mentioned in lines 306-316 in the main text. Lastly, in Poland, Chyleński et al. (2023) did not detect an increase in EEF-related ancestry from the EBA to the MBA. However, this results from the fact that the analysed material comes from the Trzciniec culture, whose roots lie largely in the forest zone of Europe. Samples from Poland — and in fact from Silesia (south-western Poland) — may differ from what Chyleński observed. Silesia is much closer, in many respects, to the regions of Bohemia and Moravia, so this should not be surprising. We have added a few sentences presenting the genetic background for the Early Bronze Age in Central Europe to the Introduction, lines 90 to 110.

For Central Germany, however, the increase in EEF ancestry from the early to the late phase of the LBA represents a new finding based on the newly reported data. From an

archaeological perspective, it does not come as a complete surprise given the degree of interconnectedness and long-term interaction among neighbouring LBA groups visible/attested in the archaeological record, as discussed specifically in lines 585-590 of the discussion for the southern parts of Germany and Bohemia. Hence, a delayed increase of EEF ancestry in Central Germany could simply be the result of geographic distance. Finally, while published LBA genomic data from Poland remain limited, the higher EEF ancestry observed in our four individuals follows the broader trend of increasing EEF ancestry in the LBA in Central Europe. Therefore, our qpAdm results agree, for most regions, with established patterns.

More generally, the observed genetic shifts during the LBA are detectable but subtle in comparison to major turnover events in preceding millennia such as the Neolithic transition or the arrival of steppe-related ancestry. To our knowledge, there is no strong prediction from the archaeological record as to which degree a particular cultural influence should be reflected in the genetic ancestry profiles of the associated LBA individuals.

The second concerns the continued theme throughout the manuscript of fluctuating Early European Farmer (EEF) related ancestry. There doesn't seem to be a consistent story told about population continuity vs admixture, but I'm sure a few sentences would clear it up. Some examples: The paper highlights a fairly consistent increase in EEF ancestry over time in all regions (Figure 3), but the qpAdm modeling suggests population continuity from EBA to early LBA (Central Germany) or continuity from MBA to LBA (Bohemia and Poland).

Although EEF ancestry increases overall across the entire Bronze Age period, the timing, extent, and mechanisms differ by region. We have clarified this in the relevant results and discussion sections to avoid implying a uniform demographic process and to emphasise the nuances the new data offers with regards to periods of stasis vs. periods of change (Fig. 3b), and in particular in light of contextual evidence from isotopic and archaeological data.

In the strontium analyses, everyone seems local, regardless of time depth, genetic sex, or mortuary practice – where is the increase in EEF ancestry coming from?

Strontium and ancient DNA analyses capture mobility at different temporal levels. Specifically, strontium isotope ratios reflect childhood residence since tooth enamel forms early in life and does not remodel. Hence, strontium isotope analysis can detect only first-generation migrants. Genetic ancestry, on the other hand, records population interactions over many generations. As a result, individuals can appear isotopically local while still carrying ancestry components introduced into the region by earlier gene flow and interaction with neighbouring groups in the interconnected Late Bronze Age world. Concerning the potential sources of the increased EEF ancestry, we refer to this in the results section of the main text, e.g. in lines 330-343 and discuss this in lines 583-590.

In the dietary analyses, there does seem to be a correlation with a return to C₃ plants and EEF-related ancestry, in the late LBA, but not with the earlier switch to millet. The authors then bounce between suggesting a change in ancestry with the former, but not one with the latter. Settlement contexts is another example.

Concerning the diet of the individuals, the association between higher EEF ancestry and the late LBA return to C₃ plants does not necessarily imply causation. This pattern may reflect changes in agricultural practices, environment, culinary preferences or cultural influence rather than population movement. Indeed, we did not observe a change in the ancestry of the individuals when millet became part of the diet in the early phase of the LBA. The same can be applied to the settlement contexts, that is, an increase in settlement contexts is not necessarily caused by a change in the ancestry of individuals. We have rearranged the relevant parts of the discussion and have now emphasised this aspect in the main text.

The authors could expand the mtDNA and NRY analyses and interpretations. The continental spread and phylogenetic history of the haplotypes typed are not discussed, even in passing. Are any of them surprising (the two CT individuals)? All expected? How about the drastically increased mitochondrial diversity? KUC023 has a unique NRY haplotype and an elevated Sr ratio, but is not apparently a genetic outlier. What might be going on there?

CT is a very basal Y haplogroup, and we could only resolve these individuals to this resolution due to poor coverage. CT simply indicates that these individuals do not carry Y haplogroups from the A and B branches (which is not surprising, as these are constrained to Africa in the human past). Aside from these calls, and aside from KUC023, we observe only common haplogroups from I1, I2, R1a, and R1b, which are expected at this time.

Individual KUC023 carries Y-haplogroup I-Z63, which is rare and, as a result, poorly understood. The earliest find of a derived haplogroup is in Denmark (McColl et al. 2024), dated to 400-220 BC. We find mostly evidence for individuals carrying similar Z63-derived sub-lineages in both Northern Europe and Austria/Hungary in the later Iron Age and Medieval periods (or even later), but this tells us little about their origin or distribution during the LBA. There is a general hypothesis that it originated in Scandinavia, as it is still found there and in the British Isles today. Hence, this may point to a more northern ancestry source in this individual's past, which may not be readily obvious genetically due to overall heterogeneity in the region at the time. This is now mentioned in the Supplementary Note 2.4.

Regarding the mitochondrial diversity, we observed 24 different mtDNA haplogroups in 28 female individuals, broadly resembling the mt haplogroup distribution in Bronze Age Central Europe (e.g., Brandt et al. 2013, Science; or Papac et al. 2021 (Unetice)). This is now mentioned in the main text.

Some quick comments on the methods:

Was aDNA extraction attempted on any of the bones from the cremated remains? Are the authors just not reporting libraries at all that didn't pass internal endogenous DNA scan requirements? They had the bones for the isotopic work.

We did not attempt DNA extraction from the cremated bones used for strontium isotope analysis, as these samples were fully calcined, therefore lacked preserved organic material. Moreover, the samples selected for isotopic work were mostly fragmented long bone fragments, which are generally less suitable for aDNA analysis. We did, however, attempt DNA extraction from one cremated petrous bone from Kuckenbug (KUC029), which did not yield usable results (very low endogenous DNA and no damage profiles). We agree that this information should be reported, and have now included the individual KUC029 Supplementary Table 1.

In your biological relatedness analyses, KIN (<https://doi.org/10.1186/s13059-023-02847-7>) seems to be growing in popularity and perhaps should be attempted here?

We have now included KIN (Popli et al. 2023) in our biological relatedness analyses under Supplementary Table 5c. The results are consistent with those obtained using READv2 and BREADR, confirming the inferred degrees of relatedness among the individuals. Details of this analysis have been added to the Methods (Biological relatedness) section.

Furthermore, is the limited relatedness detected at Kuckenbug and Esperstedt to be expected? The authors could expound on this more.

Limited close biological relatedness within cemeteries and mortuary contexts is increasingly recognised and described in prehistoric Europe. Although these examples come from different periods and regions, many studies now report low levels of close biological relatedness within mortuary groups. A recent study from Neolithic Ireland (Carlin et al. 2025), for example, shows that even long-used, well-sampled mortuary contexts often contain individuals who are only distantly related or unrelated, reflecting socially defined mortuary practices rather than biological kin groups. Similar patterns have been identified in Early Bronze Age Central Europe (Knipper et al. 2017; Mitnik et al. 2019), where cemeteries often also contain individuals of diverse origins with few close biological relatives. Another example comes from Knipper et al. 2020, which examines a 5th-century AD cemetery at Mőzs in Pannonia. Isotopic analyses revealed a population of diverse geographical origins that formed a socially constituted community rather than a biologically related one. These examples, which should be used with appropriate caution, suggest that the limited relatedness observed at Kuckenbug and Esperstedt is not that unexpected and may reflect selective mortuary traditions and community belonging. The inclusion and analysis of more individuals in future studies will refine this picture further. We have now added a paragraph to the Discussion, lines 694 - 709, that expands on this.

Specific Comments:

Page 3, lines 81-83: Give a date range here?

We have now adjusted the text to read as “*Based on recent archaeological findings, this practice is proposed to have originated in the middle Danube region, including areas of modern-day Lower Austria, Moravia, southwest Slovakia, and parts of Hungary⁶, in the period between ca. 2000 and 1600 BCE, before spreading more widely throughout Europe and becoming consolidated in the classic Urnfield horizon after 1300 BCE*”.

Page 5, lines 139-141: Are the "Sonderbestattungen" burials the only unique archaeological/cultural aspect of the Kuckenburg and Esperstedt settlements? Would the results presented here be applicable to the other non-cremation sites mentioned in Paragraph 2 of the introduction? How broadly applicable would these analyses be to Bronze Age central European communities, or do the authors want the takeaway to be a hyper site-specific perspective?

As the phenomenon of “settlement burials” (in all its varied definitions) is quite common not only in the southern parts of present-day Saxony-Anhalt and Bohemia but also across a wide range from Central Germany to present-day Moravia, the results of our analysis are not limited to a hyper-site-specific perspective. Rather, they provide a model for ritual and social activities in these regions. We have clarified this point in the text, and now it reads as “*Hence, by analysing the individuals buried at Kuckenburg and Esperstedt, we aim to address gaps in our understanding of Central European LBA communities on site-specific and regional scales, especially as forms of settlement burials are widespread across Central Germany, Bohemia, and Moravia and therefore provide a broader comparative framework*”.

Page 5, lines 162-164: I can follow why NES005 was thrown away, but I'm not following why NES006 wasn't - high contamination with angsd and hapcon, but low with ContamMix. Have you considered you might have sex chromosomal aneuploidies? Could check with karyo_RxRy? Also, if I'm following your column BW in Sup Table 1, you only threw out 2 individuals (though NES006 is in red), which seems to leave 72 individuals for other analyses; not 70. Are there duplicates? Column H has 4 "same" values, and maybe that is where the difference is. If so, mention 2 duplicates were found in 4 analyses.

Thank you for pointing this out. NES006 should indeed have been excluded. Although ContamMix produced a lower contamination estimate, both ANGSD and HapCon indicated high contamination levels. We also re-examined NES006 regarding potential sex-chromosomal aneuploidies, and the individual is confidently classified as XY with posterior probability ~1. No NES individuals were flagged as potentially carrying sex chromosomal aneuploidies. Regarding the sample numbers, four samples represent two biological individuals (as noted in Supplementary Table 1, column H “same”, and now specified in the Notes of the same table). After removing these duplicates, the dataset contains 69 unique

individuals. We have now clarified this in the text and corrected the numbers accordingly. We have also updated all relevant tests and figures to exclude NES006.

Page 7, lines 181-182: Which individuals (genomic data) are the authors using to define these three groups in their analyses? I can't find those group IDs in any of the supplementary tables. If they are consistent samples that everyone uses, define and give appropriate references.

Thank you for highlighting this issue. The three main ancestry groups used in our f_4 -statistics follow the publication of Patterson et al. (2022). We have now specified in the text how EEf, WHG, and Steppe correspond to the entries in Supplementary Table 2b, and indicated the column ("Group ID_published") where they can be found.

Page 7, lines 183-187: This belongs more in the Methods section than in Results

We agree that these sentences read more like a methods section. However, we chose to include this brief explanation in the Results to help readers interpret the described tests without needing to navigate back and forth between sections.

Page 7, lines 194-196 (Figure 2b Legend): An expanded introduction that includes the S. Germany, Czechia, and Poland individuals would help give us more perspective of how we should be interpreting the qpAdm models in comparison to the Unstrut population.

We agree that a brief review of the general genetic background of Bronze Age Central Europe was missing from the Introduction. We have added a full paragraph (lines 90-103) including the relevant literature to set the scene for the interpretation of the newly reported results of this study.

Page 8, lines 219-220: A description of what makes this cemetery unique likely belongs in the introduction.

Thank you for this suggestion. However, we believe that a short sentence explaining why the Neckarsulm cemetery is considered unique fits best at the beginning of the Results section, as it helps situate the reader for the upcoming discussion of results. We have now expanded this sentence and direct the reader to the more detailed archaeological description provided in Supplementary Note 1.4, where the site and the burials are discussed. The text now reads: "*For comparison, we also analysed 12 individuals from the unique LBA cemetery of Neckarsulm (NES) in South Germany, which has been interpreted archaeologically as the burial place of an all-male "warrior community" (Supplementary Note 1.4)*". We also reference the publications by Knöpke (2009) and Wahl & Price (2013), which offer comprehensive archaeological and isotopic interpretations of the site. In our study, we only provide new genetic data for these individuals.

Page 8, line 240: Edit to “...generated, as well as published, LBA...”

We have revised the sentence.

Page 9, lines 256-258: I don't understand where the support for this sentence comes from in the preceding paragraph, as I don't see any graphics or data in Supp Table 3 showing any early Bronze Age comparisons?

We have now added all supporting f_4 -statistics for this sentence to the text (lines 268–287) and added all the corresponding tests in Supplementary Table 3. We have also adjusted the sentence in the main text for clarity, which now reads: “*Together, the observed patterns attest to similar shifts of genetic ancestry in all regions, characterised by increasing proportions of EEF-related ancestry between the Early and the Late Bronze Age, although the timing, extent, and mechanisms of this shift differ between regions*”.

Page 9, lines 260-269: Why are the individuals from Poland, who seem to have the lowest proportion of EEF-related ancestry, not discussed in this paragraph?

We have now included the Polish individuals in this paragraph, which now reads as “*The four LBA individuals from SW/Central Poland also show a lower proportion of EEF-related ancestry (37.4 ± 2.2 %) similar to those from Central Germany. However, given the very small sample size, it is unclear whether these individuals are representative of Polish LBA groups more broadly, and therefore these results should be interpreted with caution*”.

Page 9, lines 271-274: Can you be more clear where you are getting the proportions used for Figure 3a/b, because they don't seem to exactly match the values from qpAdm listed in Figure 2b (for the LBA individuals, at least). This can be put in the text or in the figure legend.

The proportions used for Figure 3a/b are the qpAdm distal ancestry estimates presented in Supplementary Table 4a (qpADM_distal). These are the same proportions as in Figure 2b. We have clarified this in both figure captions. We have now added the EEF proportions that are used in Figure 3a/b for the published individuals in Supplementary Table 4a as well.

Pages 12-13, lines 333-342: Also discuss LNV001 and SNY001, who are highlighted as outliers in this manuscript, in the PCA, and in Figure 5 below.

They are now mentioned in this section, and the text reads as “*LNV001 and SNY001 from Bohemia show highest genetic similarity to individuals from Central Germany*”.

Page 14, lines 365-374: Why are KNE028, KUC027 not being discussed in the hapROH analysis? Do they not also have elevated levels? See Supplemental Figure 3.

We have now included KUC027 and KNE028 in the discussion of individuals with elevated ROH values. Both display moderate ROH levels (sum_roh > 4 = 83.72 cM and 83.89 cM, respectively), comparable to KUC010 (72.68 cM). In addition, we have specified the units (cM) for all ROH values in the text and added this information to the caption of Supplementary Table 7.

Page 14, lines 386-396: None of the genetic outliers appear non-local in the isotopic analyses. Can the authors discuss/highlight this?

We copy here our answer from above:

Strontium and ancient DNA analyses capture mobility at different temporal levels. Specifically, strontium isotope ratios reflect childhood residence since tooth enamel forms early in life and does not remodel. Hence, strontium isotope analysis can detect only first-generation migrants. Genetic ancestry, on the other hand, records population interactions over many generations. As a result, individuals can appear isotopically local while still carrying ancestry components introduced into the region by earlier gene flow and interaction with neighbouring groups in the interconnected Late Bronze Age world. We have added this note in Supplementary Note 3.2.

Page 15, line 406: How is millet consumption determined? I see the column in Sup Table 1, but not a description.

Millet consumption was inferred from $\delta^{13}\text{C}$ (and $\delta^{15}\text{N}$) values measured on bone collagen, following established interpretative thresholds from previous isotope studies in the region (e.g., Filipović et al., 2020; Le Huray, 2005, also see Orfanou et al., 2024, for a full explanation of the method). We set a $\delta^{13}\text{C}$ collagen value of -18‰ as the threshold for increased consumption of C_4 plants. Values higher than -18‰ indicate the consumption of a mixed C_3/C_4 diet and moderate millet intake, while values higher than -12‰ correspond to a diet predominantly based on C_4 plants (Filipović et al., 2020; Pospieszny et al., 2021). We have added this clarification to the text to specify how millet consumption was determined. The threshold note has also been added in Figure 6d above the dotted line of -18‰ .

Page 21, line 571: By “this dietary shift”, I assume the authors are referring to the return C_3 plants, but please clarify.

Indeed, this is what we mean by “this dietary shift”. We have clarified this in the text by replacing “this dietary shift” with “the return to C_3 -plant-based diets”.

Page 27, line 730: The 1240k capture needs references; I assume you aren’t using the new Twist kit.

We have now added the appropriate reference for the 1240k capture method.

Page 27, lines 733-736: Both the mtDNA and NRY capture protocols/kits need references.

We have now included the relevant references for the mitochondrial and Y chromosomes capture protocols.

Page 30, line 804: I assume these genotypes are from the preceding ATLAS calls? Please be more specific in the "Population Genetic Analyses" when you used the pseudo-haploid calls and when you used the ATLAS genotypes.

We thank the reviewer for addressing this ambiguity. A specification was added in the "Imputation" and "Population Genetic Analyses" sections. We specified that "*Diploid autosomal genotypes produced with ATLAS*" were used for imputation with GLIMPSE ("Imputation" section); we also now specify that "*All the population genetic analyses besides IBD analyses were conducted with pseudo-haploid genotypes produced with pileupCaller*" in the "Reference dataset" section, and then that regarding the dataset for IBD analyses "*Diploid genotypes imputed with GLIMPSE and with >500,000 SNPs and genotype probabilities >0.7 were included in the analysis.*" in the "IBDs" section.

Page 30, line 807: spelling "pubblication"

Corrected.

Page 31, lines 831-833: Was the HO dataset *only* used for the PCA and Admixture analyses? If nothing else, say that. I would also state the limited HO dataset was necessary to include many of the contemporary populations.

Yes, the Human Origins (HO) dataset was only used for PCA and ADMIXTURE analyses. We have clarified this in the revised text and noted that the HO dataset was chosen to enable inclusion of a larger number of modern comparative populations.

Page 31, line 853: Here and elsewhere, when you bring up population group labels like this, reference your Supplementary Table 2b.

Done.

Page 31, lines 856-857: As already previously mentioned, we are never given who is included in these groups.

The three main ancestry groups used in our f_4 -statistics follow the publication of Patterson et al. (2022). We have now specified in the text how EEF, WHG, and Steppe correspond to the entries in Supplementary Table 2b, and indicated the column ("Group ID_published") where they can be found.

Page 32, lines 867-872: Are these labels from the AADR or from labels in Supplementary Table 2b? I don't see some of them in the supplementary tables.

The labels correspond to the AADR v54.1.p1 group designations and are also provided in Supplementary Table 2b under the column "Group ID." We have now clarified this in the text by explicitly referencing Supplementary Table 2b alongside the labels.

Page 37, lines 1031-1032: I applaud the authors for including this level of detail for their statistical tests.

Thank you.

POINT BY POINT RESPONSE TO REVIEWER COMMENTS

Reviewer #1 (Remarks to the Author):

I was really pleased to see the careful and thoughtful revisions and responses to the reviewers. The authors have made an already strong paper excellent. I have no further comments - well done!

-Catherine Frieman

Thank you. Much appreciated.

Reviewer #2 (Remarks to the Author):

The authors have provided a comprehensive reply to the comments and have made improvements to their manuscript. I have only one further comment: the reply makes clear the source of the collagen isotopic data. For complete clarity I think they ought to cite Orfanou et al. (2024) as the source in the caption for Figure 6d.

Thank you very much. A reference to our dietary isotope study is now included in the caption of Figure 6.

Reviewer #3 (Remarks to the Author):

I have reviewed the manuscript for the second time, and in my opinion, the authors satisfactorily modified the text in accordance with my former suggestions. I appreciate the time and effort they have put into acknowledging these. Thus, I see no reason why the manuscript could be published.

Thank you. Much appreciated.

Reviewer #4 (Remarks to the Author):

I appreciate the authors for responding positively and meticulously to most of my suggestions. I now read a more nuanced and regional take on the EEF ancestry changes, more highlights of when results matched or didn't match expectations, and expanded discussions of the diet correlations and relatedness measurements. More specifically, thanks to the authors for running KIN, for reporting on the single cremation they attempted aDNA analysis on, for the new supplemental note on KUC023, for clarifying which genotype sets were used when, and for providing much more detail on ancestry group names and membership, both in the text and in supplemental tables – this makes future replication easier and our field better.

We agree and highly appreciate the thorough review. Many thanks!

(All subsequent line references are to the tracked changes pdf)

In regards to the individuals genotyped from South Germany, Bohemia, and SW/Central Poland, I acknowledge the broader genetic background of Bronze Age Central Europeans given in the Introduction and I acknowledge that the authors more consistently highlight these individuals in the Results. I am still unclear why, however, these individuals are only being introduced to us halfway through the Results, particularly when most of the first few paragraphs of that section

incorporate them as part of a larger Central European LBA gene pool and they are identified in Figure 1. While I understand the focus is on the Esperstedt and Kuckenburg Unstrut group, would it truly disorganize the manuscript to summarize those 3 additional populations in a minimized version of what you do in lines 132-159 for your main site(s)? You describe as much in your peer review rebuttal notes. Instead, we still only get a single sentence heads up, in the final paragraph of the introduction, of their existence, even though they represent almost half of your (newly contributed) dataset. It is the authors' prerogative to bury (ha) or to highlight sites, but I do think it is a missed opportunity. If LBA individuals from Central Europe are universally scarce, shouldn't any and all be meaningful?

We fully agree. It was and is clearly not our intention to place some sites over others. The reason why our main focus lies on Esperstedt and Kuckenburg is because the locally funded project was built around these two sites associated with the Unstrut group, including the generation and evaluation of all contextual data, such as mortuary practices, dietary and mobility isotopes, anthropology and other archaeological meta-information.

We now have introduced the co-analysed comparative sites/regions in the introduction and refer to the respective Supplementary Notes for more specific details on the sites and the inhumations studied.

Some new small edits:

In regards to KUC023, fix the reference at line 190 to say 'Supplementary Note 2.4' instead of 'Supplementary Table 2.4', and add a reference to Supplementary Note 2.4 somewhere in lines 433-436.

Thanks. This is done.

Additionally include a short note of KUC027 and KUC010 in Supplementary Note 2.3, as you have already done in the main body of the text. That is where the reader will see the actual results.

Thank you. A short note is now added to the Supplementary Note 2.3.

Thank you for clarifying on final sample counts (69 passing quality control), though your edits on lines 186-187 now says "51 different mtDNA haplotypes in 64 individuals", and I'm unclear why.

Thank you for this comment. The reason why mitochondrial haplogroups could be assigned to only 64 of the 69 individuals passing quality control is that five individuals did not have enough mitochondrial reads for reliable mtDNA haplogroup calling. This clarification has now been added to the main text, and it now reads as "We observed 51 different mt haplotypes in 64 individuals, with enough mt reads for reliable mtDNA haplogroup calling, broadly resembling the mt haplogroup distribution in Bronze Age Central Europe."

There are some changes in font style at lines 944-946 and 1084-1086.

Thanks for spotting. The font style in these lines was already fixed in the clean version, as is in the resubmitted 2nd revision.